# Characterization and Antitumoral Activity of Biohybrids Based on Turmeric and Silver/Silver Chloride Nanoparticles

**DOI:** 10.3390/ma14164726

**Published:** 2021-08-21

**Authors:** Marcela-Elisabeta Barbinta-Patrascu, Yulia Gorshkova, Camelia Ungureanu, Nicoleta Badea, Gizo Bokuchava, Andrada Lazea-Stoyanova, Mihaela Bacalum, Alexander Zhigunov, Sanja Petrovic

**Affiliations:** 1Department of Electricity, Solid-State Physics and Biophysics, Faculty of Physics, University of Bucharest, 405 Atomistilor Street, P.O. Box MG-11, 077125 Bucharest-Măgurele, Romania; marcela.barbinta@unibuc.ro; 2Frank Laboratory of Neutron Physics, Joint Institute for Nuclear Research, Joliot-Curie, 6 Dubna, 141980 Moscow, Russia; gizo.bokuchava@jinr.ru; 3Institute of Physics, Kazan Federal University, 16a Kremlyovskaya Street, 420008 Kazan, Russia; 4General Chemistry Department, Faculty of Applied Chemistry and Materials Science, University “Politehnica” of Bucharest, 1-7, Polizu Street, 011061 Bucharest, Romania; nicoleta.badea@upb.ro; 5Low Temperature Plasma Department, National Institute for Lasers, Plasma and Radiation Physics, 409 Atomistilor Street, Magurele, 077125 Ilfov, Romania; andrada@infim.ro; 6Department of Life and Environmental Physics, “Horia Hulubei” National Institute of Physics and Nuclear Engineering, 077125 Măgurele, Romania; bmihaela@nipne.ro; 7Institute of Macromolecular Chemistry AS CR, Heyrovskeho nam. 2, 162 06 Prague 6, Czech Republic; zhigunov@imc.cas.cz; 8Department of Chemical Technology, Faculty of Technology, University of Nis, Bulevar Oslobodjenja 124, 16000 Leskovac, Serbia; milenkovic_sanja@yahoo.com

**Keywords:** biomimetic membranes, “green” silver/silver chloride nanoparticles, turmeric (*Curcuma longa* L.), biohybrids, antioxidant activity, antibacterial action, antitumoral activity

## Abstract

The phyto-development of nanomaterials is one of the main challenges for scientists today, as it offers unusual properties and multifunctionality. The originality of our paper lies in the study of new materials based on biomimicking lipid bilayers loaded with chlorophyll, chitosan, and turmeric-generated nano-silver/silver chloride particles. These materials showed a good free radical scavenging capacity between 76.25 and 93.26% (in vitro tested through chemiluminescence method) and a good antimicrobial activity against *Enterococcus faecalis* bacterium (IZ > 10 mm). The anticancer activity of our developed bio-based materials was investigated against two cancer cell lines (human colorectal adenocarcinoma cells HT-29, and human liver carcinoma cells HepG2) and compared to one healthy cell line (human fibroblast BJ cell line). Cell viability was evaluated for all prepared materials after a 24 h treatment and was used to select the biohybrid with the highest therapeutic index (TI); additionally, the hemolytic activity of the samples was also evaluated. Finally, we investigated the morphological changes induced by the developed materials against the cell lines studied. Biophysical studies on these materials were done by correlating UV–Vis and FTIR absorption spectroscopy, with XRD, SANS, and SAXS methods, and with information provided by microscopic techniques (AFM, SEM/EDS). In conclusion, these “green” developed hybrid systems are an important alternative in cancer treatment, and against health problems associated with drug-resistant infections.

## 1. Introduction

Nano size particles formed from a reduced chemical element keep the properties of the used element as the main characteristics. The large surface area of metal nanoparticles (MNPs) increases their interaction with other molecules. The advanced properties of MNPs allow nanoparticles to be applied in various fields such as diagnostics, pharmaceutics, and therapy. The most used nanoparticles are platinum, silver, palladium, gold, copper, cadmium, titanium, and zinc oxide and cadmium sulphide nanoparticles [1]. Due to their good antimicrobial activity, silver nanoparticles (AgNPs) have been used in medicine, stomatology, textile, cosmetic, agriculture, and pharmaceutical industries [2,3]. Silver has been used as antibacterial agent against many microorganisms, but a large spectrum of new results about AgNPs’ use as antibacterial and antiviral agents [2,4,5,6,7] show that AgNPs have not only antibacterial but proven antiviral activity, as well to HIV-1 virus [8] and hepatitis b type [9].

Production of nanoparticles might be taken by physical or chemical methods [1,10]. Over the recent years, green nanotechnology for the synthesis of MNPs that employs various reducing and stabilizing agents-phytochemicals is considered to be the simplest and most cost-effective method. Green synthesis of AgNPs is considered to have less ecological risks [6].

During the last decade, the “green” revolution in all the fields shifted to plant kingdom utilization. Phyto-mediated synthesis of MNPs is a very promising area in Green Nanotechnology, as it involves economic, simple, fast, and ecological strategies, the vegetal extracts having a dual role: they act both as a reducing agent, as well as a capping agent for metal nanoparticles formation. Extracts obtained from plants or waste plants material contain ketones, aldehydes, proteins, flavones, etc., that play a role of reducing and capping agents in the formed AgNPs [11]. The “green” AgNPs method, in which reducing molecules from extracts binds at the particles’ surface, can be considered successful when a change of the nanoparticles’ suspension color occurs [12]. Extracts, most often generated from plant leaves of blackberry, raspberry [12], banana [13], pepper [14], *Desmodium* plant [15], etc., have been used as a source of phytochemicals for green AgNPs synthesis method, but also from other plant parts, seeds, root, or stem.

“Green” synthesis of AgNPs is an eco-friendly and pollution-free process for obtaining a significant source of bioactive principles for medicine, pharmacognosy, and pharmaceutical industries. “Green”-AgNPs possess good antibacterial activities thanks to the bio-compounds on their surface that come from the precursor plant extract. Thus, AgNPs phyto-synthesized from leaves aqueous extracts of *Osmanthus fragrans* [16], *Nageia*
*nagi* [17], or *Vitis vinifera* [18] exhibited a good biocidal effect against both Gram-negative and Gram-positive bacteria, having high potential for biomedical applications. A valuable and appreciated plant is *Curcuma longa* L. (turmeric) which has numerous biological activities, such as antiviral, anticancer, antimicrobial activities, antirheumatic, hepatoprotective, anti-inflammatory, and antioxidant properties [19].

Aqueous extract from *Curcuma longa* leaf is used as a reducing and capping agent in silver nanoparticles formulations [20,21,22,23,24]. Numerous studies have shown a synergistic effect between AgNPs and curcumin. The antimicrobial mechanism of the AgNPs nanocomposites attached to microbial surface is expressed by high Ag^+^ concentration near the surface of microorganism, but an increase in Ag^+^ release in the presence of curcumin generates more reactive oxygen species (ROS), causing bacterial death [25]. The antimicrobial property of AgNPs is governed by shape, stability, aggregation state, medium, capping agent type, as well as methods of surface functionalization of nanoparticles. The most widespread antimicrobial efficacies of AgNPs can be conditioned by truncated or spherical AgNPs form [19]. However, spherical AgNPs are usually preferred. Additionally, a skin disease that is influenced by lack of pigments due to an absence of melanocytes and leukoderma can be treated by AgNPs from *Curcuma longa* of 46.60 nm and spherical shape [26].

The use of AgNPs from *Curcuma longa* L. obtained by simple reduction also found use as nano-fluorescent agent used for improving medical diagnostics [27]. The formulated AgNPs from *Curcuma longa* L. coated cotton fabrics may be utilized for a variety of applications in hospital patients, such as a wound healing agent [19].

The main objective of our research work was to obtain turmeric-generated silver/silver chloride nanoparticles (Ag/AgCl NPs) and their use in nano-formulation of antitumoral biohybrids containing chitosan and chlorophyll-loaded biomimetic membranes. Our study is the first report on the phyto-synthesis of Ag/AgClNPs from turmeric rhizome extract in a biomimetic fluid (phosphate-buffered saline solution, PBS pH 7.4).

Ag/AgCl plasmonic materials have biocidal and antioxidant properties, and low cytotoxicity (as compared to AgNPs) assured through the Ag^+^ solubility equilibrium controlled by the AgCl, which allows for a low level of silver ions to be released into the environment [28,29,30].

Veronica da Silva Ferreira et al. reported on the antitumor potential of AgCl and Ag/AgCl nanoparticles against the Ewing’s sarcoma cells [31].

Due to the good biological performance of such plasmonic hybrids, in our study, we developed a simple method for synthesis of hybrid Ag/AgClNPs from turmeric rhizome extract in the presence of physiological phosphate-buffered saline solution (PBS) compared with fabricated AgNPs using turmeric extracts in water medium [32] or curcumin in alkaline medium [33].

The obtained biohybrids, containing Ag/AgClNPs, chlorophyll-loaded biomimetic membranes, and chitosan, were spectrally, structural, and morphological characterized, and they were investigated for their antioxidant activity, antimicrobial activity (against the clinical pathogen *Enterococcus faecalis*) and antitumoral effect.

We chose the bacterium *Enterococcus faecalis* because from a bacterium that affects only the dental root, lately it has started to be more frequently present in intrahospital infections [34]. *Enterococcus faecalis* (Enterococci family) is an important cause of nosocomial infections and intrahospital infections, due to vancomycin resistant Enterococci (VRE) occurring frequently. A high number of cases in prevalence of VRE has been reported in the last years in many regions. Enterococci are the second most frequent cause of bacteremia, nosocomial urinary tract infection, infective endocarditis, gastrointestinal, intra-abdominal, pelvic, and oral infections. *Enterococcus faecalis* has become an important microorganism in public health because of multidrug resistance in VRE [35].

## 2. Materials and Methods

### 2.1. Materials

Silver nitrate (AgNO_3_), Potassium dihydrogen phosphate (KH_2_PO_4_), disodium hydrogen phosphate (Na_2_HPO_4_), peptone, Tris (hydroxymethylaminomethane base), hydrochloric acid (HCl), hydrogen peroxide (H_2_O_2_), and luminol (5-amino-2,3-dihydro-phthalazine-1,4-dione) were purchased from Merck (Darmstadt, Germany). Sodium chloride (NaCl), soybean lecithin, and chitosan (CS, from crab shells, highly viscous, ≥85% deacetylated) were supplied from Sigma Aldrich (Darmstadt, Germany). Silver behenate (CH_3_(CH_2_)_20_COOAg) powder used in SAXS study was purchased from Alfa Aesar, a Johnson Matthey Company.

Chlorophyll *a* (Chla) was extracted in our laboratory from fresh spinach leaves according to Strain and Svec’s method [36]. Turmeric (*Curcuma longa* L.) rhizomes were acquired from a local market.

Antibacterial assay of the samples was evaluated against pathogenic Gram-positive bacterium, *Enterococcus faecalis* ATCC 29212. All materials used for assay were purchased from VWR (Darmstadt, Germany). *Enterococcus faecalis* were sub-cultured onto Luria Bertani Agar acc. Miller (LBA) plates at 37 °C.

Cell culture and reagents: For this study, one healthy cell line and two cancer cell lines were used for the in vitro studies. Human fibroblast BJ cells (ATCC CRL-2522, Manassas, VA, USA) and human colorectal adenocarcinoma HT-29 cells (ATCC, Manassas, VA, USA) were grown in Minimum Essential Medium (MEM) supplemented with 2 mM L-Glutamine, 10% fetal calf serum (FCS), 100 units/mL of penicillin, and 100 µg/mL of streptomycin at 37 °C in a humidified incubator under an atmosphere containing 5% CO_2_. HepG2 cells (ATCC, Manassas, VA, USA) were grown in Dulbecco’s Modified Eagle Medium (DMEM) supplemented with 2 mM L-Glutamine, 10% fetal calf serum (FCS), and 100 units/mL of penicillin and 100 µg/mL. All cell cultivation media were purchased from Biochrom AG (Berlin, Germany). Drabkin reagent and standard hemoglobin were purchased from Sigma Aldrich (Darmstadt, Germany).

### 2.2. Green Synthesis of Biohybrids Based on Turmeric-Generated Ag/AgClNPs

#### 2.2.1. Preparation of Biomimetic Membranes

Biomimetic membranes labelled with chlorophyll *a* (Chla), without and with chitosan (CTS) coating, were prepared by the hydration of soybean lecithin thin film as previously described [37]. In the artificial lipid bilayers, Chla was added in a percentage of 1.2% (w/w) related to the lipid content. The lipid vesicles without and with CTS (named Lip and Lip-CTS, respectively) were suspended in a simulating bio-fluid: the phosphate-buffered saline solution (PBS, KH_2_PO_4_/Na_2_HPO_4_/NaCl, pH 7.4).

#### 2.2.2. Preparation of Silver/Silver Chloride Nanoparticles

A mixture of 20 g of *Curcuma longa* L. rhizome powder and 200 mL of distilled water was boiled for 5 min, in a covered 400 mL beaker. After cooling, this mixture was filtered, and the obtained filtrate represents the turmeric aqueous extract (Tur) which was kept and stored at −20 °C until further use.

A volume of 1 mL of Tur extract was mixed with a volume of 100 mL of 1 mM AgNO_3_ aqueous solution under continuous magnetic stirring, and then placed at room temperature for one day. The resulting suspension was diluted with PBS pH 7.4, in a volume ratio of 8:5 (80 mL of Ag/AgClNPs was mixed with 50 mL of PBS pH 7.4), and furthermore subjected to an ultrasound treatment in an ultrasonic bath (BRANSON 1210, Marshall Scientific, Hampton, NH, USA) for 30 min. The resulting suspension represents the sample Tur-nAg/AgCl that should be stored in the refrigerator in the dark.

#### 2.2.3. Eco-Design of Biohybrids

Three types of biohybrids with chitosan and/or biomimetic membranes were achieved by “green” bottom-up approach described in [37].

Briefly, different mixtures of: (Tur-nAg/AgCl + Lip + PBS), (Tur-nAg/AgCl + CTS + PBS), and (Tur-nAg/AgCl + Lip–CTS + PBS) were subjected to strong stirring (VIBRAX stirrer, Milian, OH, USA, 200 rpm) for 15 min followed by ultrasound treatment in an ultrasonic bath (BRANSON 1210, Marshall Scientific, Hampton, NH, USA) for 30 min, resulting in the biohybrids: Tur-nAg/AgCl–Lip, Tur-nAg/AgCl–CTS, and Tur-nAg/AgCl–Lip–CTS, respectively, with a final lipid concentration of 0.34 mg/mL, the final silver content of 0.61 mM, and 0.01% (*w*/*v*) of CTS.

### 2.3. Physico-Chemical and Biological Characterization of Developed Materials

#### 2.3.1. Spectral, Structural, and Morphological Characterization

The UV–Vis absorption spectra of the samples were recorded on a double-beam Jasco V-670 UV–vis–NIR spectrophotometer (Jasco, Tokyo, Japan), from 200 to 800 nm (at the resolution of 1 nm).

The identification of functional groups and examination of type of interaction that occurs between the components of biohybrid complexes were performed using Fourier-transform infrared (FTIR) spectroscopy. The FTIR spectra were registered on an Excalibur FTS–3000 FTIR spectrometer (Bio-Rad, Munich, Germany) at a temperature of 40 ± 1 °C. The experimental data were collected in the wavenumber range of 4000–500 cm^−1^ and at a spectral resolution of 0.4 cm^−1^. Each spectrum was averaged over 128 scans, then for all spectra, the baseline correction and normalization were applied.

Atomic force microscopy (AFM) images were recorded in tapping mode at room temperature on NTEGRA PRIMA module (NT-MDT Spectrum Instruments, Zelenograd, Russia). The samples were sonicated to avoid the formation of aggregates, and 50 μL aliquots were immediately dropped onto freshly cleaved mica substrate (15 mm × 15 mm). Each complex was scanned after 3 h of air drying at room temperature with a scan rate of 0.3–0.5 Hz, using commercial NSG01 tips of a 10 nm curvature radius. Final height profiles were obtained using the NT-MDT Spectrum Instruments Image Analysis P9 software.

Zeta potential was determined, in triplicate, via electrophoretic mobility of the particles in an electric field, on Malvern Zetasizer Nano ZS (Malvern Instruments Inc., Worcestershire, UK) at 25 °C, and the mean values were reported.

A scanning electron microscope (SEM) and a FEI Inspect model S50 apparatus (Hillsboro, OR, USA) were used to analyze the surface morphology of the samples. The images presented in this study are secondary electrons images, obtained at a 10 kV acceleration voltage. The magnifications ranged between 50× and 20,000×, for a working distance of 10 mm. In order to avoid surface charging effects and prior to the SEM investigations, an approximate 10 nm Au layer was used to cover the sample’s surfaces. A sputtering Cressington 108 auto sputter coater apparatus, equipped with a Cressington mtm 20 thickness controller, was used to obtain the desired Au thin film. Moreover, the aforementioned SEM apparatus was also equipped with an Energy-dispersive X-ray spectroscopy detector (EDAX, Berwyn, PA, USA) used to obtain the EDS spectra.

The X-ray diffraction (XRD) was used for examination of crystallographic structure and phase composition of the silver-based samples. XRD spectra were acquired during a 12 h exposure for each sample on EMPYREAN diffractometer (PANalytical, Almelo, The Netherlands) with Cu-Kα incident radiation having the wavelength λ = 1.5406 Å. The average crystallite size (*D*) was calculated according to Scherrer’s equation:(1)$$D=\frac{K\lambda }{\beta \cos\theta }$$
where *K* is a dimensionless shape factor close to unity, θ is the Bragg angle, and β is the peak width at half the maximum intensity (FWHM). The instrumental line broadening was estimated before FWHM calculation.

Small-angle scattering (SAS) was applied for structural characterization of the components of the biohybrids.

The size and shape of hybrid Ag/AgClNPs in excess PBS was extracted using Small angle X-ray scattering (SAXS). Despite the low concentration solutions were used, experiments became successful due to high contrast between silver and solvent, as intensity $I\left(Q\right)\;\sim \;({\bar{\rho }}_{e}-\rho_{s,e}$). The estimated values of scattering length density (SLDe) for a given SAXS setup are as follows: ρ_Ag,e_ = 77.9 × 10^−6^ Å^−2^, ρ_s,e_ = 9.47 × 10^−6^ Å^−2^. In addition, the size and shape of Tur-nAg/AgCl in the analysis of complex systems (Tur-nAg/AgCl–Lip, Tur-nAg/AgCl–CTS, and Tur-nAg/AgCl–Lip–CTS) can also be easily obtained, because the SAXS results are less sensitive to the other components of the investigated systems due to the much lower contrast. The estimated SLDe values for chitosan (ρ_CTS,e_) can be taken in range of 1.36 × 10^−6^–2.73 × 10^−6^ Å^−2^ for density of (C_6_H_11_NO_4_)_n_ in range of 0.15–0.3 g/cm^3^ and calculated SLD values of the soybean lecithin liposomes (ρ_Lip,e_ ≈ 9.4 × 10^−6^ Å^−2^) is very close to the SLDe of H_2_O. Additionally, we can neglect the contribution of the scattering on the liposomes due the extremely low concentration (C_Lip_ = 0.33 mg/mL) in our case. SAXS experiments were carried out on an instrument with using a pinhole camera (MolMet, Rigaku, Japan, modified by SAXSLAB/Xenocs, Grenoble, France) attached to a microfocused X-ray beam generator (Rigaku MicroMax 003) with operating parameters 50 kV and 0.6 mA (30 W). The camera was equipped with a vacuum version of Pilatus 300 K detector. The position of the center of the beam line and the sample-detector distance were calibrated using a standard sample of silver behenate [38]. The samples were placed in borosilicate capillaries with an internal diameter of 2 mm and a wall thickness of 0.01 mm (W. Muller, Berlin, Germany). X-ray wavelength of λ = 1.54 Å was used in the experiment. The SAXS data were collected in the range of momentum transfer Q =  (4π/λ) sin(θ/2), where θ is the scattering angle, from 0.005 to 0.5 Å^−1^. Data reduction was carried out using homemade software based on PyFAI Python library [39]. We have used a generalized Guinier-Porod model [40], as was described in detail in [37] for extracting the radius of gyration (Rg), shape parameter (s), and fractal dimension (Dm) of the silver/silver chloride nanoparticles.

The structure organization of liposomes was investigated using Small-Angle Neutron Scattering (SANS). For this purpose, a suspension of multilamellar lipid vesicles (MLVs) was prepared in biological buffer PBS suspended in D_2_O (NRC “Kurchatov Institute”-PNPI, Gatchina, Russia). The measurements were performed at YuMO spectrometer at the IBR-2 pulsed reactor (Dubna, Moscow region, Russia). The time-of-flight spectrometer was in two-detector configuration [41]. The data were collected in the q-range of 0.007–0.4 Å^−1^. The solutions of the biohybrid nanocomplexes were placed in quartz cells with internal thickness of 1 mm (Hellma, Germany). The measurements were done at 37 ± 0.2 °C. The SANS spectrum exposition time was 10 min for each sample. The initial experimental data were processed using the SAS program [42], where the background was subtracted [43]. The repeat distance d of the multilamellar liposomes was determined by fitting the diffraction peak by the Gaussian function.

Then, d was calculated from the position of the maximum of the diffraction peak Q_max_ as d = 2π/Q_max_. Analysis of the SANS and SAXS spectra was performed using SasView software [SasView. Available online: http://www.sasview.org/ (accessed on 18 July 2017)].

#### 2.3.2. In Vitro Antioxidant Activity

The scavenger activity of free oxygen radicals by turmeric-derived samples was evaluated using the chemiluminescence systems based on luminol (0.01 mM) and H_2_O_2_ (0.01 mM) in TRIS-HCl buffer solution pH 8.6. in vitro antioxidant activity was calculated using the following equation:AA = [(I_0_ − I)/I_0_] ∙ 100%, (2)
where I_0_ is the maximum CL intensity at t = 5 s for the control, and I is the maximum CL intensity for each sample at t = 5 s [44].

All the samples were analyzed in triplicate using the Turner Design TD 20/20 USA Chemiluminometer.

#### 2.3.3. Antibacterial Assay of Tested Samples

To evaluate the antibacterial activity, we used the agar well diffusion method [45,46,47]. Petri dishes were inoculated with *Enterococcus faecalis* bacterium over the entire LBA surface. Then, a hole with a diameter of 6 mm is punched aseptically with a sterile Durham tube, and a volume of 50 μL of the antimicrobial agent is introduced into the well. Then, LBA plates are incubated under suitable conditions at 37 °C for 24 h. The antibacterial agent diffuses in the LBA and inhibits the growth of the bacterial strain tested, then the diameters of inhibition growth zones are measured (IGZ, mm).

#### 2.3.4. Cell Viability

The effect of biohybrids on the cells’ viability was determined using MTT (3-(4,5-dimethylthiazol-2-yl)-2,5-diphenyltetrazolium bromide) assay, as described previously [48]. First, the cells were seeded in 96 well plates and further cultured for 24 h. Following, the samples were added into the medium for 24 h. As negative control we used cells grown only in medium. After one day, the medium was discarded and 1 mg/mL MTT solution was added to each well and incubated for additional 4 h at 37 °C. Finally, the solution was discarded, and the insoluble formazan product was dissolved in DMSO. Finally, the samples absorbance was recorded using a plate reader Mithras 940 (Berthold, Germany) at 570 nm. All values were corrected for the background by subtracting the blank and cell viability was obtained using the following equation: [(A_570_ of treated cells)/(A_570_ of untreated cells)] × 100%. The sample’s concentration that was able to reduce cell viability by half (IC_50_) was extracted by fitting the experimental data with a logistical sigmoidal equation in Origin 8.1 software (Microcal Inc., Los Angeles, CA, USA).

#### 2.3.5. Morphological Evaluation of Cells

Cells were grown on glass cover slips and treated with two concentrations of nanoparticles for 24 h. Then, the cells were washed with Phosphate-buffered saline (PBS), then fixed with 3.7% formaldehyde dissolved in PBS for 15 min, washed with PBS, stained for 15 min with 20 μg/mL acridine orange (Sigma-Aldrich, Darmstadt, Germany), and finally washed in PBS. Finally, the images were recorded using an Andor DSD2 Confocal Unit (Andor, Ireland), mounted on an epifluorescence microscope, Olympus BX-51 (Olympus, Germany), and equipped with a 40× objective and an appropriate filter cube (excitation filter 466/40 nm, dichroic mirror 488 nm, and emission filter 525/54 nm).

#### 2.3.6. Hemocompatibility

The new hybrids hemocompatibility was investigated as reported in ASTM F 756-00 standard [48]. Blood from healthy volunteers was collected and diluted to a final hemoglobin concentration of ~10 mg/mL with PBS. Following that, the blood was incubated for 4 h with the highest concentration of the samples at 37 °C and kept under constant agitation. At the end, the blood was centrifuged, and the supernatant was collected and mixed in volume report of 1:1 with Drabkin reagent (Sigma-Aldrich). The solution was left to react for 15 min, then the absorbance was recorded at 570 nm using Mithras 940 plate reader. Human red blood cells in PBS and distilled water were used as negative and positive controls, respectively. The recorded values were corrected for dilution factors and background, and used to calculate the hemolytic index (i.e., percentage of hemolysis) using the following equation:% Hemolysis = (A_S_/A_T_)100%(3)
where A_S_ represents the corrected absorbance of the hemoglobin released in supernatant after treatment with the samples and A_T_ represents the corrected absorbance of the total released hemoglobin.

### 2.4. Statistical Analysis

All data were expressed as (mean value ± standard deviation) of three individual experiments. Statistical analysis was performed using *Student’s test* (Microsoft Excel 2010) to determine the significant differences among the experimental groups, and the *p*-values < 0.05 were considered statistically significant.

## 3. Results and Discussion

### 3.1. Optical Characterization of “Green” Developed Materials

In order to get useful information regarding the biohybrid formation, the obtained materials were optically characterized by UV–Vis absorption and FTIR spectroscopy.

UV–Vis absorption spectra displayed in Figure 1 revealed the spectral signature of silver nanoparticles (409 nm) and of chlorophyll (667 nm) in the biohybrids. Other peaks in the UV region are assigned to the carbohydrates and/or peptide bonds in proteins (225–235 nm), polyphenols, or aromatic amino acid residues of proteins (259–261 nm) arising from vegetal extract [49,50]. The chitosan coating led to the appearance of more pronounced peaks (217; 251; 264 nm), due to the interaction between the amino and hydroxyl groups of chitosan and the functional groups of biocompounds derived from turmeric extract.

The formation of biohybrids was further confirmed by FTIR analysis (see Figure 2). Thus, the medium sharp band located at 1355 cm^−1^ in Tur-nAg/AgCl split into two bands: 1351/1370 cm^−1^ in Tur-nAg/AgCl–Lip, and 1357/1373 cm^−1^ in Tur-nAg/AgCl–CTS. For Tur-nAg/AgCl–Lip–CTS, this band weakened and shifted to 1375 cm^−1^. These FTIR bands are assigned to the vibrations of –OH, –CH groups arising from phenols [51] of vegetal extract.

The medium band at 1075 cm^−1^ in Tur-nAg/AgCl, attributed to stretching vibration of –C–O–C– groups of polysaccharides (arising from plant extract) and to the symmetric PO_2_^−^ stretching (arising from PBS) [48,52], shifted to 1087 cm^−1^ in Tur-nAg/AgCl–Lip, 1072 cm^−1^ in Tur-nAg/AgCl–CTS, and to 1088 cm^−1^ (strong and sharp band) in Tur-nAg/AgCl–Lip–CTS.

The band at 1744 cm^−1^ in Tur-nAg/AgCl, corresponding to –C=O stretching of esterified carboxylic groups (–COOCH_3_) [53], slightly moved to 1740 cm^−1^ in Tur-nAg/AgCl–CTS, 1741 cm^−1^ in Tur-nAg/AgCl–Lip, and to 1738 cm^−1^ (very strong and sharp band) in Tur-nAg/AgCl–Lip–CTS.

In all the samples, we identified the bands at 2855 cm^−1^ (attributed to C−H symmetrical stretch vibration of alkyl chains [54] and at 2925 cm^−1^ (referring to the aldehydic C−H anti-symmetric stretching mode [55]). These bands are sharp and strong in Tur-nAg/AgCl–Lip–CTS.

It could be observed the very broad FTIR bands centered at 3313 cm^−1^ for the samples: Tur-nAg/AgCl, Tur-nAg/AgCl–Lip, and Tur-nAg/AgCl–CTS, attributed to stretching vibrations of hydroxyl groups H-bonded in polysaccharides, alcohols, and phenolic compounds (e.g., curcumin arising from turmeric extract) and to N–H stretching vibrations [48]; these functional groups belong to bio-compound cocktails of turmeric extract. In Tur-nAg/AgCl–Lip–CTS, this band is very intense and sharp and is centered at 3389 cm^−1^.

Other significant changes in FTIR band position/intensities occurred in the following wavenumber ranges:800–1000 cm^−1^ assigned to aromatic rings of curcumin arising from turmeric extract, and to saccharide structure [56].1000–1100 cm^−1^ attributed to vibrations of the –C–O–C– and –C–O–H bonds present in polysaccharide structures [57].1500–1700 cm^−1^ characteristic for carboxylate groups (–COO–) [51] and to amide I, arising due to carbonyl (–C=O) stretch in proteins [58].

The FTIR results complete the UV-Vis absorption data, proving the involvement of polyphenols, proteins, and polysaccharides in the formation of biohybrids.

Moreover, FTIR spectra demonstrated an interaction between the components of biohybrid systems.

### 3.2. Structural and Morphological Characterization of the Silver-Based Biohybrids

#### 3.2.1. Small-Angle Neutron and X-ray Scattering

The SANS technique was used for examination of structure organization of the liposomes. All SANS curves in Figure 3 (left) have a diffraction peak in a region of *Q*-value about 1 nm. It means that the prepared liposomes have a multilamellar structure with repeat distance *d* ≈ 6.4 nm for all studied systems. The SAXS experimental data are shown in Figure 3 (right). The parameters obtained as a result of the approximation procedure for the experimental SAXS data are collected in Table 1. The proposed Guinier-Porod model allows determination of size, shape, and dimensionality of the silver/silver chloride nanoparticles from the SAXS experiment. Scattering curves for Tur-nAg/AgCl and Tur-nAg/AgCl–Lip exhibit two structural levels. The obtained radiuses of gyration are 10.8 and 12.9 nm for small nanoparticles (first structural level), and 44.7 and 48.1 nm for large nanoparticles (second structural level) for pure silver nanoparticles and silver nanoparticles adsorbed on liposomes, respectively. The contribution of the smaller particles is less pronounced when Tur-nAg/AgCl or Tur-nAg/AgCl, with liposomes surrounded by chitosan. Nevertheless, R_g1_ was identified as 11.6 nm for Tur-nAg/AgCl–CTS and 11.1 nm for Tur-nAg/AgCl–Lip–CTS, and these values are very close to the previous one for Tur-nAg/AgCl. The identical situation was observed for the second structural level for Tur-nAg/AgCl–CTS and Tur-nAg/AgCl–Lip–CTS biohybrid complexes, when in both cases R_g2_ is near 44.7 nm. Then, the average diameter can be calculated as D_SAXS,*i*_ = 2(5/3)^1/2^R_g,*i*_ as a first approximation. The obtained values are presented in Table 1.

#### 3.2.2. X-ray Diffraction

The XRD experiment was applied to determine the dimensions of silver crystals. For this purpose, the liquid sediments were separated from the PBS solution by centrifugation at 4 °C, placed on quartz substrate, and evaporated in a vacuum chamber at room temperature. The XRD spectra for systems containing the AgNPs phytogenerated from turmeric are shown in Figure 4. The presence of Ag, which has a face-centered cubic (FCC) structure with Fm$\bar{3}.$ m space group and lattice parameter *a*_Ag_ = 0.40895 nm, is characteristic of all the obtained XRD spectra

The AgCl phase is also observed. It has a face-centered cubic (FCC) structure with Fm$\bar{3}.$ m space group and lattice parameter *a*_AgCl_ = 0.55487 nm. However, we are inclined to believe that this is not due to the initial fabrication of AgNPs, but to the formation of silver chloride crystals precisely during the interaction of silver cations with chlorine anions from the PBS buffer. For instance, Alsammarraie et al. and Khan et al. reported about only AgNPs production during “green” synthesis of silver nanoparticles using turmeric extracts in water medium [32] or curcumin in alkaline medium (1M NaOH solution) [33], respectively.

The other two phases, NaCl and sodium hydrogen phosphate hydrate (SHPH), are the remaining byproducts from PBS buffer. The NaCl phase has a face-centered cubic (FCC) structure with Fm$\bar{3}.$ m space group and lattice parameter *a*_NaCl_ = 0.5640 nm. The SHPH phase with chemical formula Na_2_HPO_3_(H_2_O)_5_ has an orthorhombic structure (space group Pmn2_1_) with lattice parameters *a* = 0.7170 nm, *b* = 0.6360 nm, and *c* = 0.9070 nm [59]. The average crystallite sizes of silver/silver chloride nanoparticles (D_XRD_) were calculated from diffraction line broadening using the Scherrer Equation (1), and are collected in Table 1.

Based on the calculated parameters D_SAXS,1_ and D_XRD_Ag_, it can be unambiguously asserted that the formed Tur-nAg/AgCl are single crystals of silver. Moreover, these nanoparticles have a slightly elongated shape (s_1_ ≈ 1), while the larger nanoparticles are shaped like a sphere (s_1_ ≈ 0) with a diffusive interface (m_2_ > 4) [60]. It can be assumed that larger nanoparticles are aggregates formed from smaller nanoparticles and, on average, consist of four silver crystals or are formed from single AgCl crystals (D_SAXS,2_ ≈ D_XRD_AgCl_).

#### 3.2.3. AFM Analysis of the Silver-Based Materials

To determine the size and shape of the systems under study, two microscopic methods, AFM and SEM, were used.

The AFM topologies for silver-based materials are illustrated in Figure 5. It is clearly seen from scan images with a size of 1 × 1 µm^2^ (Figure 5(A-1,B-1)) that the pure Tur-nAg/AgCl and Tur-nAg/AgCl coated with chitosan have a slightly elongated shape, as was predicted from SAXS data analysis. The average sizes of the silver nanoparticles, calculated from size distribution (Figure 5(A-2,B-2)), are 28.4 nm with standard derivation (SD) of 8.9 nm for silver nanoparticles and 38.9 nm (SD = 11.9 nm) for chitosan coated silver nanoparticles. The AFM topologies of the biohybrid complexes, including liposomes, are presented in Figure 5(C-1,D-1). The two types of the Ag/AgCl NPs were detected from image with size of 3 × 3 µm^2^ for Tur-nAg/AgCl–Lip system: unbound (i.e., “free”) and bound to liposomes. The average sizes of “free” Tur-nAg/AgCl were calculated as follows: 34.4 nm (SD = 19.0 nm) and 32.4 nm (SD = 11.4 nm) for Ag NPs–Lip system without (Tur-nAg/AgCl–Lip) and with chitosan (Tur-nAg/AgCl–Lip–CTS), respectively. The liposomes are ellipsoidal in both cases. A similar morphological reorganization was observed in [37], when pure spherical liposomes were transformed into an ellipsoidal shape in the case of the association of Ag/AgCl NPs obtained from an extract of a mixture of nettle and grape with liposomes. The size of liposomes (their long axis) is in the range of 150–370 nm for the Tur-nAg/AgCl–Lip complex. The average size of liposomes in the Tur-nAg/AgCl–Lip–CTS complex is about 132 nm.

#### 3.2.4. SEM/EDS Investigations of Silver-Containing Materials

In Figure 6, SEM images and EDS spectra are presented, respectively. As a general remark, in all images one can observe micrometric rectangular-shape aggregates. These aggregates are the salt crystals from PBS buffer which are formed during the sample preparation procedures. Besides the micrometric features, nanometric spherical particles are seen for sample Tur-nAg/AgCl (see Figure 6a) and sample Tur-nAg/AgCl-Lip (see Figure 6b), with average diameters of 165 nm and 289 nm, respectively.

As concerning EDS spectra, see Figure 6b–h, one can observe the peaks attributed to various elements: C_Kα_, O_Kα_, Na_Kα_, Si_Kα_, P_Kα_, Au_M_, Cl_Kα_, Ag_Lα_, and K_Kα_. Gold is present due to sample preparation, whereas silver is among the identified elements within the samples, hence EDS investigations are confirming the findings presented by XRD analysis (Figure 4).

### 3.3. Evaluation of Zeta Potential of the Silver-Based Particles

The surface charge of particles, a key factor for establishing the physical stability of colloidal systems, was evaluated by zeta potential (ξ) measurements (Figure 7). Turmeric-generated silver/silver chloride nanoparticles presented good physical stability (ξ = −28.03 ± 2.49 mV). Their dispersion in biomimetic membranes also resulted in stable systems (ξ = −28.53 ± 1.55 mV). The negative surfaces of these particles (Tur-nAg/AgCl and Tur-nAg/AgCl–Lip) facilitated the attachment of chitosan. The addition of chitosan shifted the zeta potential from negative values to positive ones, demonstrating the presence of the chitosan coating on the surface of these particles: ξ_Tur-nAg/AgCl–CTS_ = +14.17 ± 1.46 mV and ξ_Tur-nAg/AgCl–Lip–CTS_ = +10.40 ± 0.80 mV.

### 3.4. Evaluation of Antioxidant and Antibacterial Activities of Bio-Designed Materials

The silver-based materials developed in our study present good antioxidant properties, with antioxidant activity values ranging from 76.25 and 93.22% (see Figure 8). The most effective free radical scavenger was Tur-nAg/AgCl–CTS (AA = 93.22 ± 0.04%).

The anti-pathogenic activity of our developed turmeric-based materials was tested against *Enterococcus faecalis* bacterium.

In the last decade, there has been an increase in drug-resistant *Enterococcus faecalis* [34]. Nowadays, many antibiotics do not work against infections caused by these strains. *Enterococcus faecalis* bacterium do not usually cause problems, but people with rudimentary health conditions or a weakened immune system are more likely to contract a disease. These infections are often spread in hospitals and cause different types of infections, such as endocarditis, meningitis, bacteremia, wound infections, and urinary tract infections [61,62,63].

Figure 9 displays the biocidal properties of turmeric-generated materials against *Enterococcus faecalis.* The most potent materials against this pathogen proved to be the chitosan-based systems: Tur-nAg/AgCl–CTS (IGZ = 13 ± 0.32 mm) and Tur-nAg/AgCl–Lip–CTS (IGZ = 12 ± 0.41 mm).

AgNPs could increase the permeability of the bacterial membrane, with denaturation of bacterial proteins, and finally release silver ions inside the bacterial cell [64,65].

Additionally, turmeric (*Curcuma longa* L.) is known to exhibit antibacterial effect against a wide range of bacterial types. *E. faecalis* (Gram-positive bacteria) show a significantly higher sensitivity to turmeric than the Gram-negative ones [66].

Neelakandan and collaborators tested effectiveness of curcumin against *Enterococcus faecalis* biofilm, and the results show the use of turmeric in endodontics might prove to be advantageous [67]. Suvarna and coworkers also studied antibacterial activity of turmeric against *Enterococcus faecalis* bacterium and found that turmeric had a significant antibacterial effect against *Enterococcus faecalis* [68].

In the literature we did not find, for comparison, biohybrids with the same composition as those in this study.

### 3.5. Evaluation of Antitumoral Properties of Turmeric-Generated Materials

We investigated the effect induced by the obtained hybrids against 3 cell lines: one normal cell line (BJ cells) and two tumoral cell lines (HT-29 and HepG2). Figure 10 presents the viability curves obtained for all silver-based materials against the cells investigated. We see that for all four materials up to 25.6 mg/mL, cell viability is affected in a similar manner, independent of the cell type. For Tur-nAg/AgCl (Figure 10A), cell viability decreases down to 80 % at 25.6 mg/mL but decreases drastically when the concentrations doubles (51.1 mg/mL). Tur-nAg/AgCl–Lip compound also affected cell viability, but in a monotonous manner with increasing concentrations (Figure 10B). For this biohybrid the healthy BJ cells are less affected than HT-29 and HepG2, at the same concentrations used.

In Figure 10C the viability data are presented for Tur-nAg/AgCl–CTS against the three cell lines. The results are similar to the ones reported for Tur-nAg/AgCl. For Tur-nAg/AgCl–Lip–CTS we found that the healthy BJ cells are less affected than cancer cells at the same concentrations used (Figure 10D). In order to better assess the effects, we also calculated the half inhibitory concentration (IC_50_) for all experimental conditions and used this to calculate the therapeutic index (TI) against the healthy BJ cells (see Table 2). Tur-nAg/AgCl–Lip and Tur-nAg/AgCl–Lip–CTS were the only biohybrids active against HT-29 and HepG2 cells. The other samples did not show any specificity for the healthy or cancerous cells.

Based on these results, we can speculate that the presence of Liposomes reduces the cytotoxicity for healthy cells with increased efficiency against tumor cells.

Furthermore, we investigated the hemolytic activity against hRBCs (Figure 11). The test was performed at the highest concentration used during the viability studies (102.2 mg/mL). Tur-nAg/AgCl–Lip–CTS has no hemolytic effect at the concentration tested, while the other three biohybrids are only slightly hemolytic at the same concentration used.

The addition of chitosan and of chlorophyll-loaded biomimicking lipid bilayers to Tur-nAg/AgCl resulted in a better antiproliferative activity against cancerous cells, while the healthy ones remain unaffected or less affected (see Figure 10).

Based on the viability and hemolytic studies, we can say that Tur-nAg/AgCl–Lip–CTS can be further considered for studies with the aim of finding improved drugs with anticancer properties and reduced systemic toxicity.

Finally, we evaluated the cell morphology of all cell lines (Figure 12, Figure 13 and Figure 14).

In Figure 12, BJ cells are presented. For control cells and the cells treated at 12.8 mg/mL, we can see the specific morphology: elongated bipolar shape of the cells. When 51.1 mg/mL was applied, for all silver-based samples, we see that the morphology of the cells is affected. Cells shape is altered, the cell body and branches are smaller, and the cells tend to round up more. This is correlated with the viability test, where we found that the viability of the cells decreases to values between 40–20% depending on the biohybrids tested.

In Figure 13, the HepG2 cells imaged for similar conditions are presented. In this case, we can also see that the control cells and the cells treated at the smaller concentrations have a similar morphology, with cells gathered in clusters. However, when the concentrations increase, the cells show a different behavior, fewer cells are found in clusters, and the cells tend to spread more on the surface.

Finally, for HT-29 cells we see a similar pattern (Figure 14). When untreated or treated HT-29 cells with small concentrations of the silver-based samples, these cells are clustered, and their morphology was not affected. By increasing the concentrations, we see the cells are found individually and their size is reduced, proving that the silver-based materials affect them, as shown in the viability studies.

## 4. Conclusions

*Curcuma longa* L. rhizome aqueous extract was used as a nanogenerator of Ag/AgCl NPs, which were used to eco-design bio-active hybrid entities with chitosan and/or artificial cell membranes. Three types of biohybrids were achieved by “green” bottom-up approach.

Optical investigations by UV–Vis absorption and FTIR spectroscopy corroborated with zeta potential measurements were used to check the biohybrid formation. These analyses showed that biocompounds from plant extract, that contain NH_2_, –COOH, –COOCH_3_, –C=O, –C–O–C–, –C–O–H, –CH_2_, and –OH groups, play key roles in the formation of our “green” developed biohybrids.

Morphological (AFM, SEM/EDS) and structural (SAXS, SANS, and XRD) analyses revealed the nano-scaled dimensions of these bio-based materials, and the presence of silver/silver chloride nanoparticles in our developed materials.

These entities presented good antioxidant activities, with values ranging between 76.25 and 93.26%.

The bio-impact of the obtained biohybrids was studied in terms of antimicrobial, hemolytic, and antiproliferative activities.

The biohybrids tested have an antimicrobial effect on a pathogenic Gram-positive bacterium (*Enterococcus faecalis*) that has recently produced many infections in humans. This is probably due to a synergistic effect between all components of biohybrids that have separate antimicrobial activity.

Only the bio-based materials containing biomimicking lipid bilayers proved to be the most effective against the cancer cells investigated.

From the in vitro studies, we can say that the Tur-nAg/AgCl–Lip–CTS is the best composite, with good efficiency against cancer cells (HT-29 and HepG2) and no hemolytic activity.

## Figures and Tables

**Figure 1 materials-14-04726-f001:**
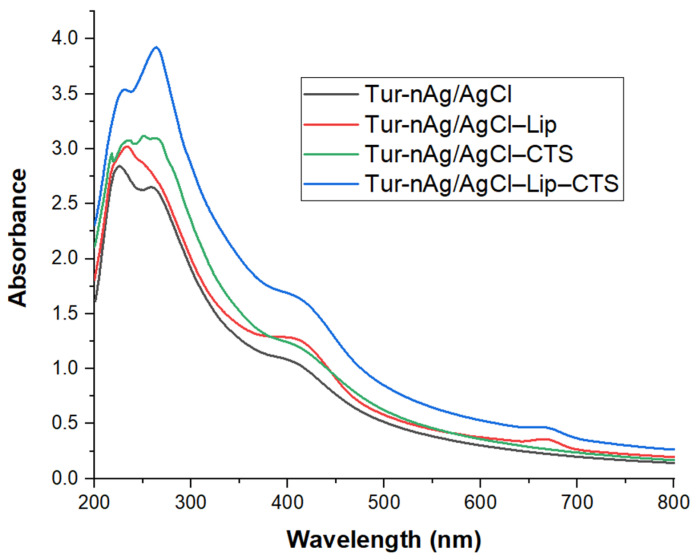
UV-Vis absorption spectra of silver-containing materials.

**Figure 2 materials-14-04726-f002:**
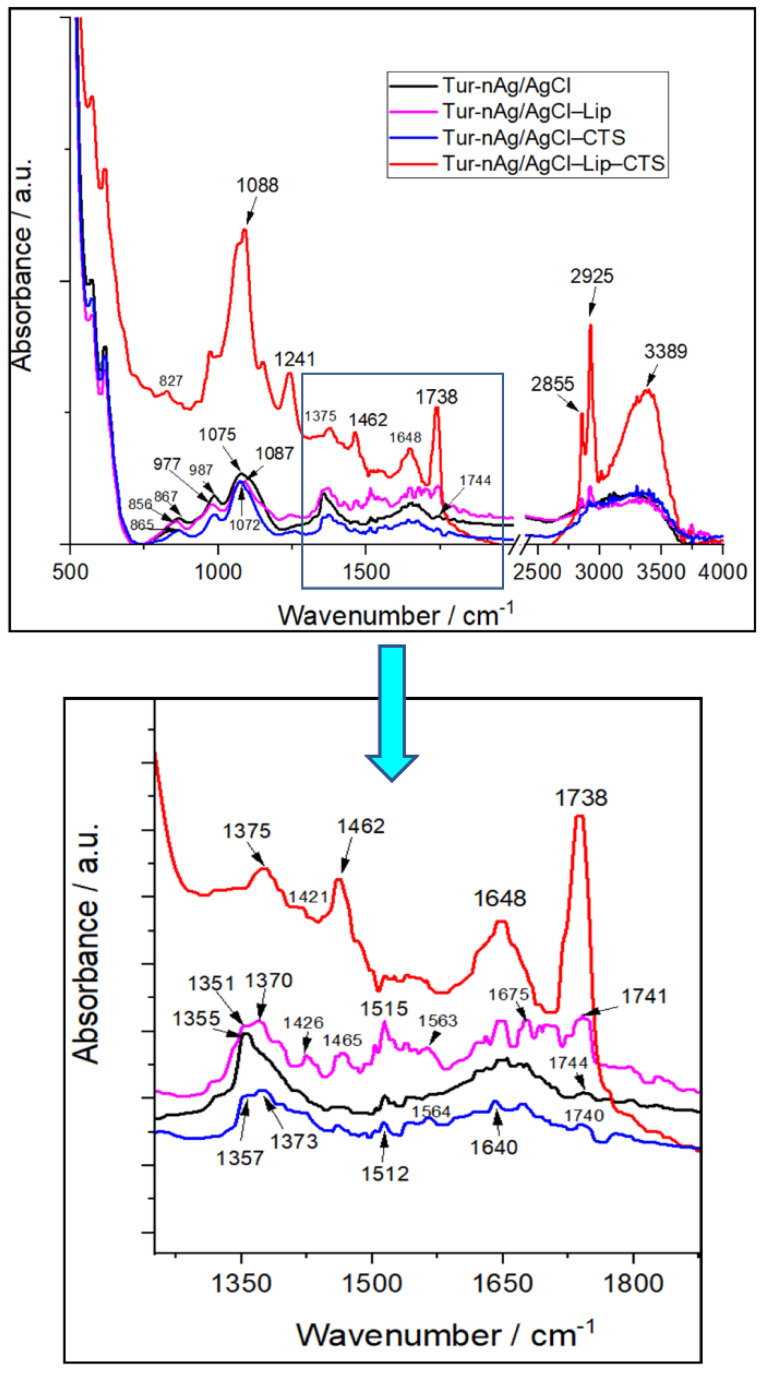
Fourier Transform Infrared (FTIR) spectra of materials generated from turmeric extract.

**Figure 3 materials-14-04726-f003:**
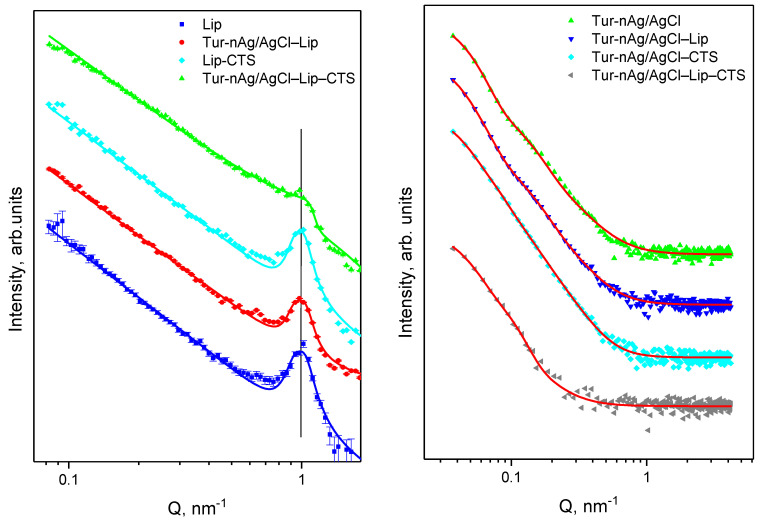
SANS (**left**) and SAXS (**right**) curves for samples in excess PBS medium. Symbols are experimental data and lines are fits. For better clarity, the curves are spaced relative to each other by a factor of 10 on the intensity scale from bottom to top.

**Figure 4 materials-14-04726-f004:**
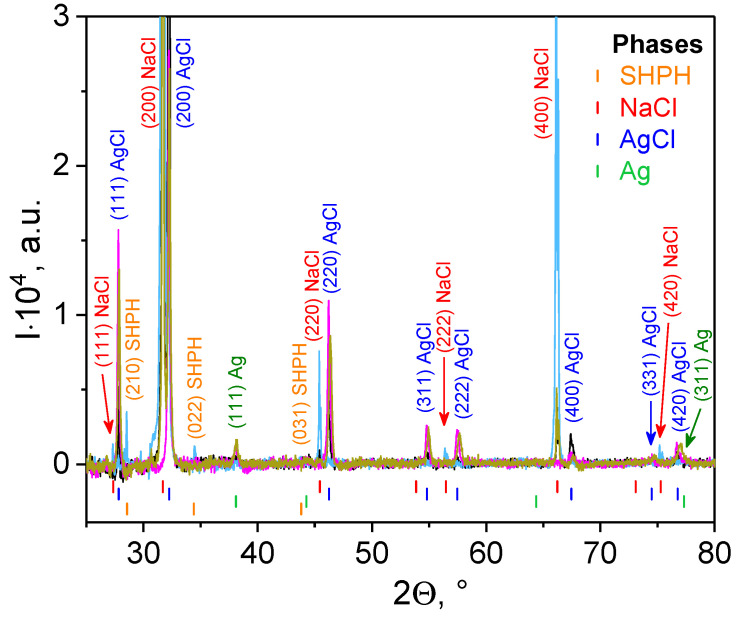
XRD patterns for samples with silver-based nanoparticles phyto-generated from turmeric: Tur-nAg/AgCl (black line), Tur-nAg/AgCl–Lip (light blue line), Tur-nAg/AgCl–CTS (magenta line), and Tur-nAg/AgCl–Lip–CTS (dark yellow line).

**Figure 5 materials-14-04726-f005:**
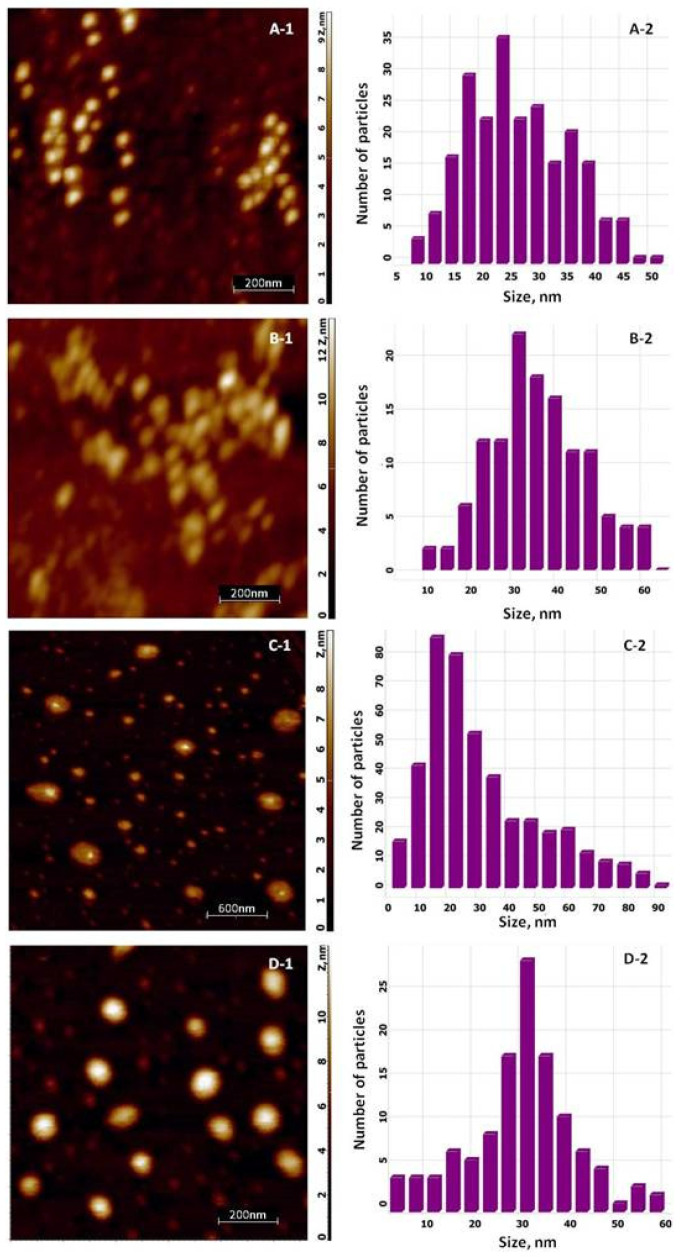
AFM images (**left**) and size distribution of NPs (**right**) for systems: (**A**) pure Tur-nAg/AgCl, (**B**) Tur-nAg/AgCl–CTS, (**C**) Tur-nAg/AgCl–Lip, and (**D**) Tur-nAg/AgCl–Lip–CTS.

**Figure 6 materials-14-04726-f006:**
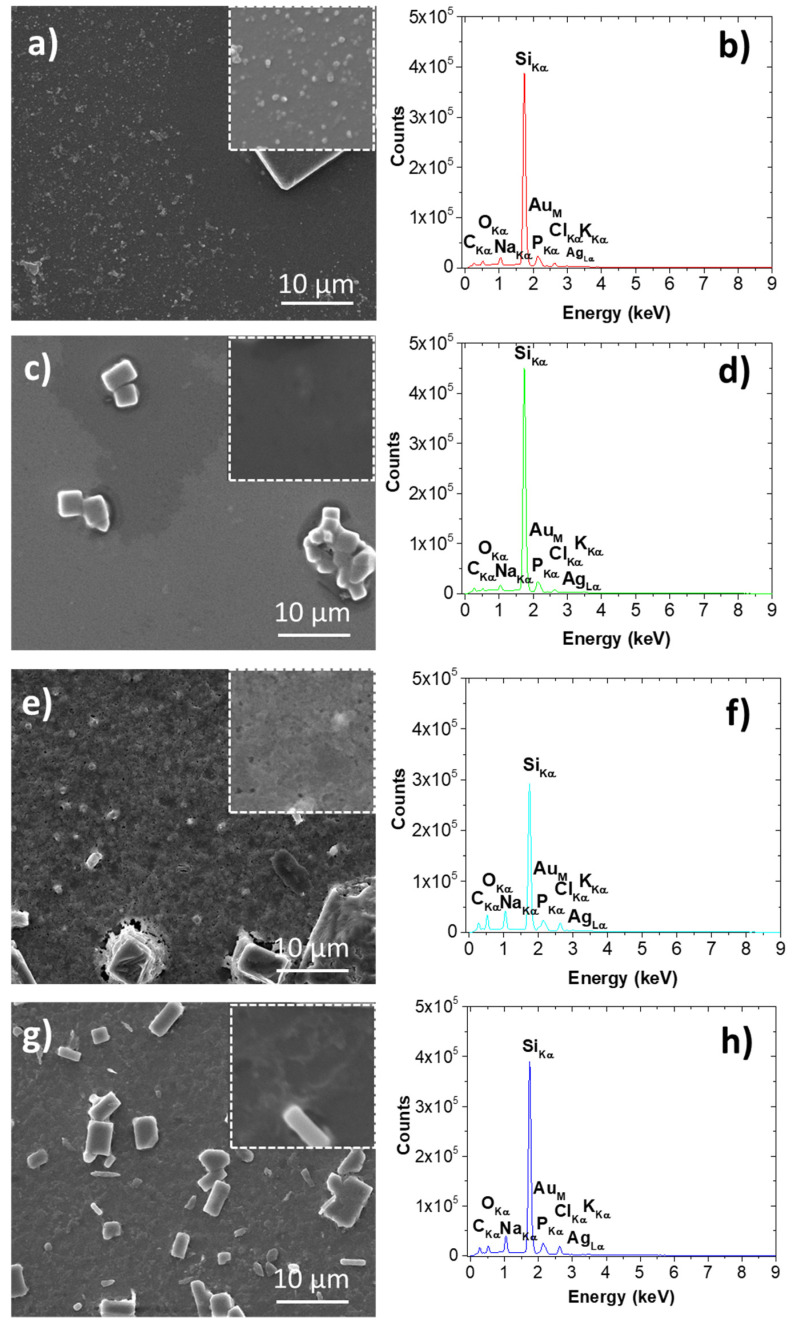
Left column presents SEM results (**a**,**c**,**e**,**g**) and right column shows EDS results (spectra **b**,**d**,**f**,**h**) for samples, from top to bottom: Tur-nAg/AgCl, Tur-nAg/AgCl-Lip, Tur-nAg/AgCl-CTS and Tur-nAg/AgCl-Lip-CTS. Magnified SEM images of 5 × 5 µm^2^ are also presented.

**Figure 7 materials-14-04726-f007:**
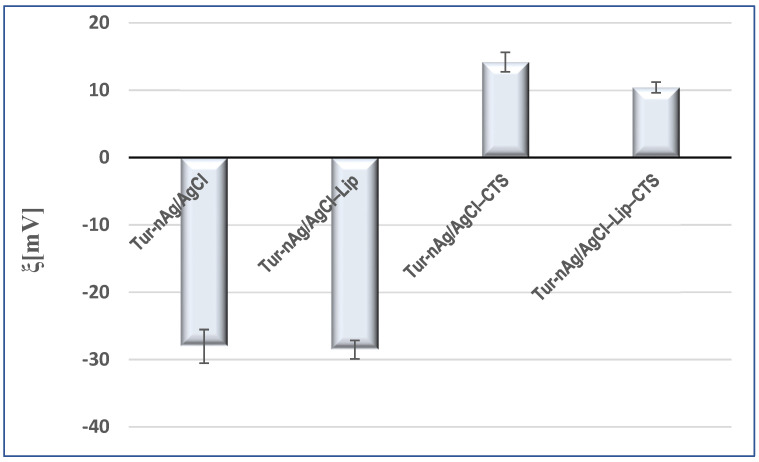
Zeta potential values of developed silver-based particles.

**Figure 8 materials-14-04726-f008:**
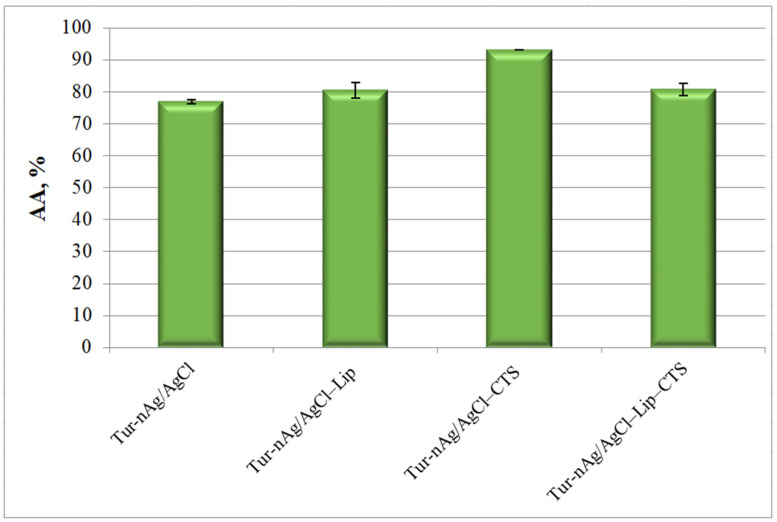
The antioxidant activity of silver-based materials.

**Figure 9 materials-14-04726-f009:**
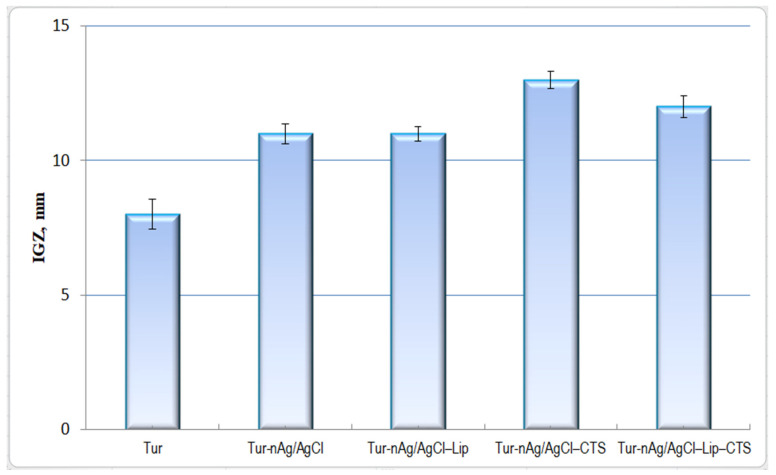
Antibacterial effect of developed materials against *Enterococcus faecalis*.

**Figure 10 materials-14-04726-f010:**
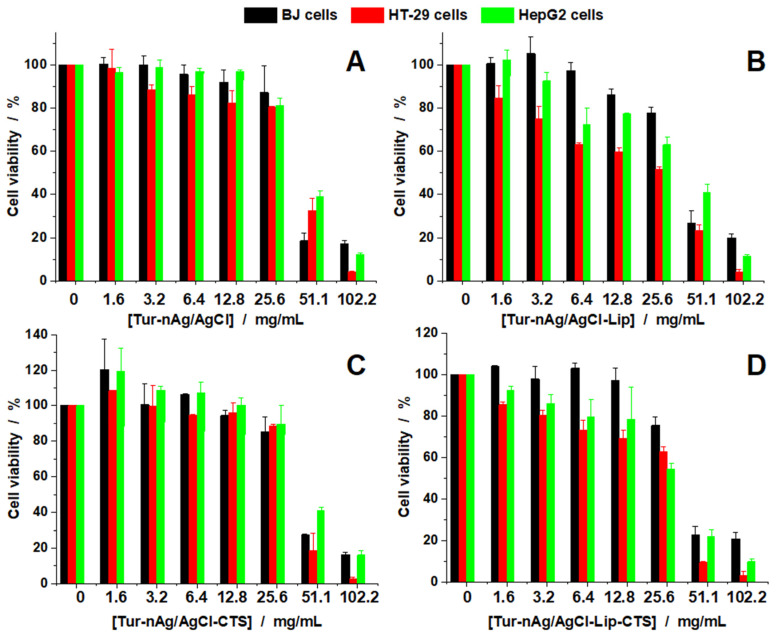
Cell viability of BJ, HT-29, and HepG2 cell lines treated with different concentrations of the samples investigated: Tur-nAg/AgCl (**A**), Tur-nAg/AgCl-Lip (**B**), Tur-nAg/AgCl-CTS (**C**) and Tur-nAg/AgCl-Lip-CTS (**D**).

**Figure 11 materials-14-04726-f011:**
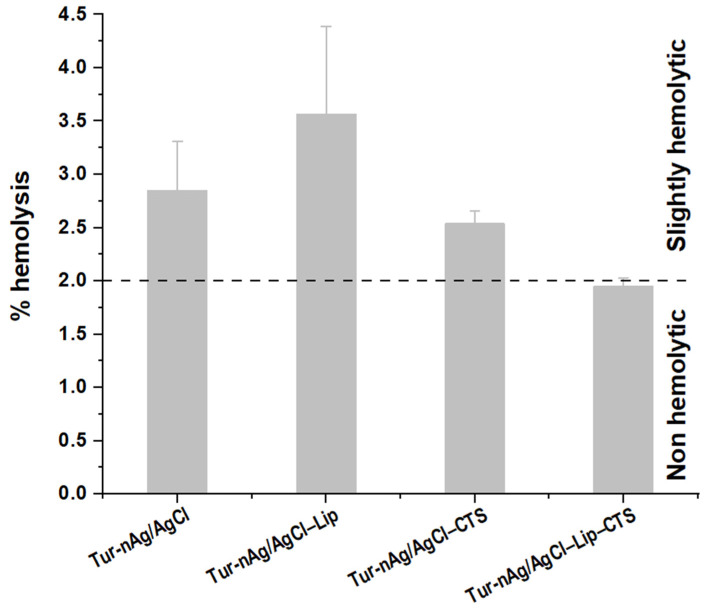
Percentage of hemolysis induced in hRBCs by the samples investigated.

**Figure 12 materials-14-04726-f012:**
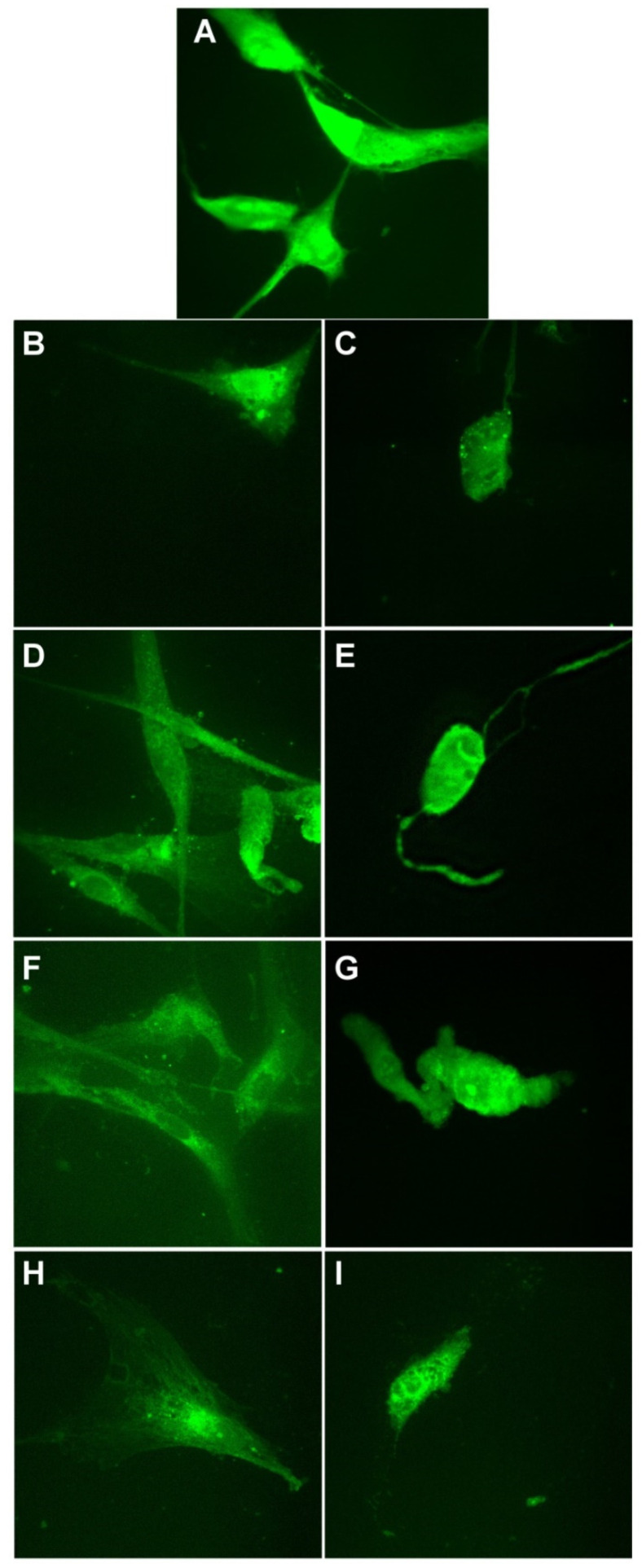
Morphological evaluation of BJ cells treated with Tur-nAg/AgCl (**B**,**C**), Tur-nAg/AgCl–Lip (**D**,**E**), Tur-nAg/AgCl–CTS (**F**,**G**), and Tur-nAg/AgCl–Lip–CTS (**H**,**I**) at two different concentrations: 12.8 mg/mL and 51.1 mg/mL, respectively, compared with control cells (**A**).

**Figure 13 materials-14-04726-f013:**
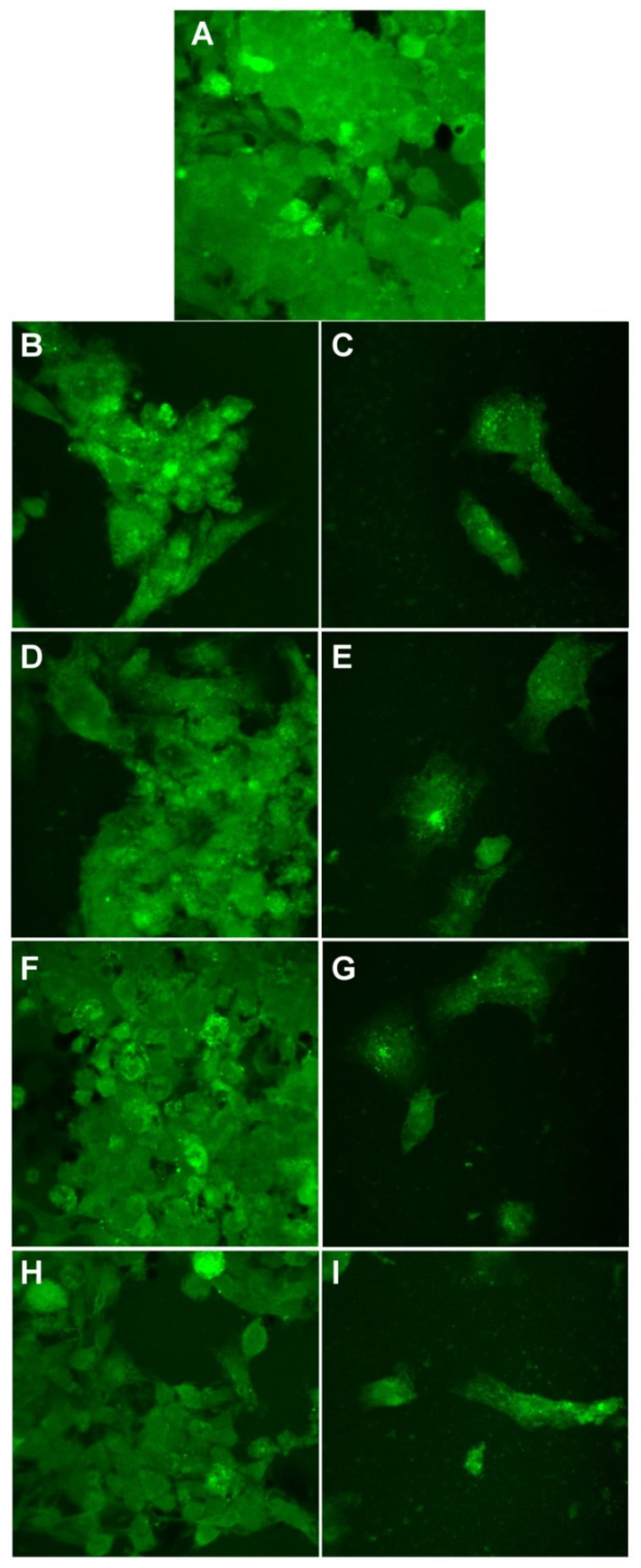
Morphological evaluation of HepG2 cells treated with Tur-nAg/AgCl (**B**,**C**), Tur-nAg/AgCl–Lip (**D**,**E**), Tur-nAg/AgCl–CTS (**F**,**G**), and Tur-nAg/AgCl–Lip–CTS (**H**,**I**) at two different concentrations: 12.8 mg/mL and 51.1 mg/mL, respectively, compared with control cells (**A**).

**Figure 14 materials-14-04726-f014:**
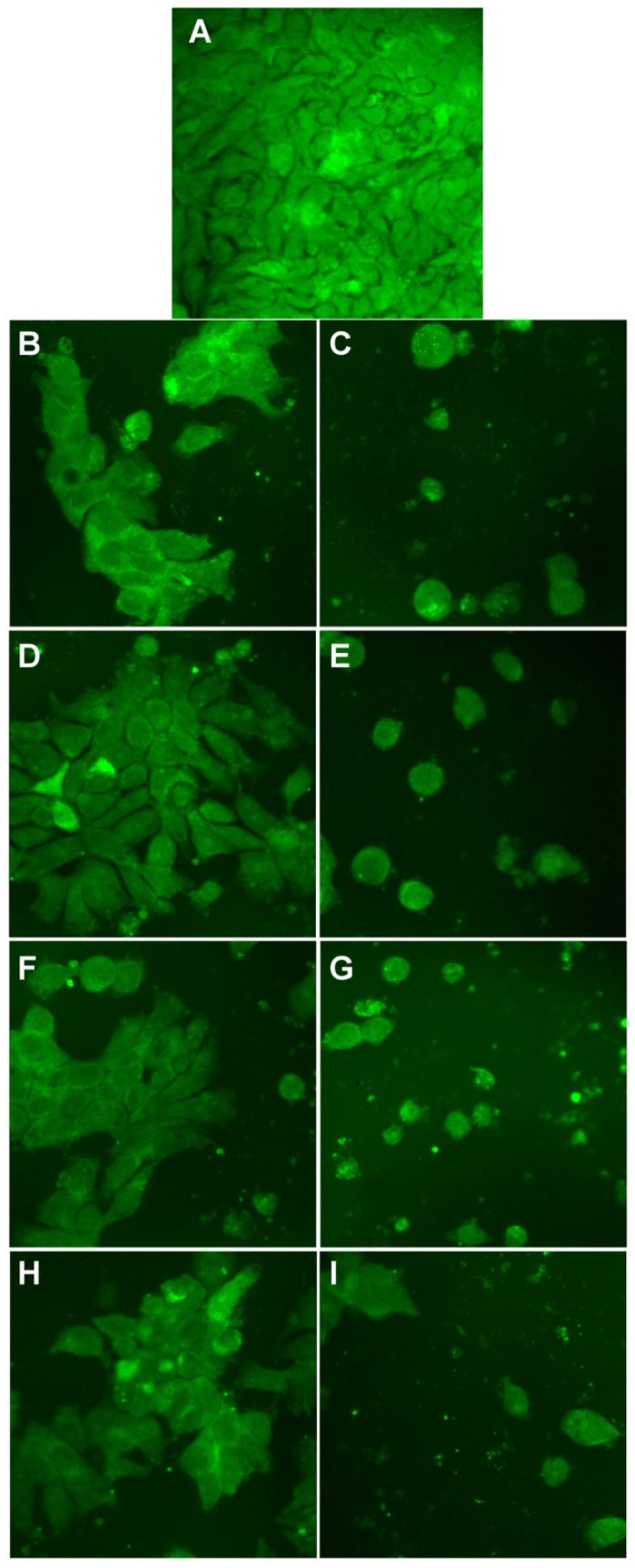
Morphological evaluation of HT-29 cells treated with Tur-nAg/AgCl (**B**,**C**), Tur-nAg/AgCl–Lip (**D**,**E**), Tur-nAg/AgCl–CTS (**F**,**G**), and Tur-nAg/AgCl–Lip–CTS (**H**,**I**) at two different concentrations: 12.8 mg/mL and 51.1 mg/mL, respectively, compared with control cells (**A**).

**Table 1 materials-14-04726-t001:** Structural parameters derived from the analysis of SAXS and XRD data of the silver/silver chloride nanoparticles for biohybrid complexes. The D_XRD_ is the average crystallite size and D_SAXS, *i*_ = 2(5/3)^1/2^R_g, *i*_ is an average diameter of Tur-nAg/AgCl.

Biohybrid Complex	Rg_2_, nm	S_2_	m_2_	Rg_1_, nm	s_1_	m_1_	D_SAXS,2_, nm	D_SAXS,1_, nm	D_XRD_AgCl_, nm	D_XRD_Ag_, nm
Tur-nAg/AgCl	44.7	0.2	4.8	10.8	1.2	3.6	115.3	27.8	102	24
Tur-nAg/AgCl–Lip	48.1	0.2	4.7	12.9	0.8	3.6	124.1	33.3	113	34
Tur-nAg/AgCl–CTS	44.8	0	4.4	11.6	1.3	3.9	115.6	29.9	97	26
Tur-nAg/AgCl–Lip–CTS	44.6	0	4.2	11.1	1.4	3.9	115.1	28.6	95	25

**Table 2 materials-14-04726-t002:** Half inhibitory concentration (IC_50_) and therapeutic index (TI) for the compounds.

Samples	IC_50_/mg/mL			TI	
	BJ	HT-29	HepG2	HT-29	HepG2
**Tur-nAg/AgCl**	36.41	41.53	44.6	0.88	0.82
**Tur-nAg/AgCl–Lip**	36.93	20.97	36.52	**1.76**	**1.01**
**Tur-nAg/AgCl–CTS**	39.28	39.69	46.03	0.99	0.85
**Tur-nAg/AgCl–Lip–CTS**	33.04	28.03	27.72	**1.18**	**1.19**

## Data Availability

The data were included in the text.

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
