# Peer review of "Characterization and Antitumoral Activity of Biohybrids Based on Turmeric and Silver/Silver Chloride Nanoparticles"

_materials, 2021, doi:10.3390/ma14164726_

Round 1

Reviewer 1 Report

1. Marcela-elisabeta barbinta-patrascu et al. have used biomimicking lipid bilayer loaded with chlorophyll, chitosan and turmeric-generated nano-silver/silver chloride particle for versatile therapeutic application, but how is this system better than silver nanomaterials alone has not been demonstrated. Hence, the rationale behind this work is not clear.

2. It is the most important to note that more detailed method how to make turmeric-generated nano-silver/silver chloride particle.

3. Why has turmeric been used in this study? There are lots of plants or biomaterials having reducing potential enough to make nanomaterials.

Reviewer 2 Report

The authors have carried a biosynthesis of the AgNPs from natural extracts, the results are discussed and presented well. My comments are below;

  1. The article should define a proper aim, now it seems a hit-and-trial work, it should focus on one purpose. If that is to biosynthesize AgNPs for anti-cancer activities, then it should discuss more on the toxicity of AgNP and other nanoparticles.
  2. The literature lacks detailed work of AgNPs synthesis from plants, the authors should add more references with extract names and their microbial activity strength too, if possible. Following articles can be added;
    1. Fabrication of Alginate Fibers Loaded with Silver Nanoparticles Biosynthesized via Dolcetto Grape Leaves (Vitis vinifera cv.): Morphological, Antimicrobial Characterization and In Vitro Release Studies
    2. Photo-irradiation based biosynthesis of silver nanoparticles by using an ever-green shrub and its antibacterial study
    3. Solar irradiation and Nageia nagi extract assisted rapid synthesis of silver nanoparticles and their antibacterial activity
  3. Can the authors comment on the scale up such AgNPs production? Their limitations too.
  4. The conclusion part should be exclusive to the main achievements of the work. Future work can be added in separate section, if authors find it obligatory.
  5. There are too many auxiliary sentences throughout manuscript.

Round 2

Reviewer 2 Report

I have no further comments to add